# The Impact of Dietary Sugars and Saturated Fats on Body and Liver Fat in a Healthcare Worker Population

**DOI:** 10.3390/nu17081328

**Published:** 2025-04-11

**Authors:** Sophia Eugenia Martinez-Vazquez, Ashuin Kammar-García, Carlos Moctezuma-Velázquez, Javier Mancilla-Galindo, Ignacio García-Juárez, Luis Federico Uscanga-Domínguez

**Affiliations:** 1Department of Gastroenterology, Instituto Nacional de Ciencias Médicas y Nutrición Salvador Zubirán, Mexico City 14080, Mexico; drinter77@gmail.com (I.G.-J.); luis.uscangad@incmnsz.mx (L.F.U.-D.); 2Dirección de Investigación, Instituto Nacional de Geriatría, Mexico City 10200, Mexico; akammar@inger.gob.mx; 3Department of Medicine, Division of Gastroenterology, University of Alberta, Edmonton, AB T6G 2X8, Canada; moctezum@ualberta.ca; 4Institute for Risk Assessment Sciences, Utrecht University, 3584 CS Utrecht, The Netherlands; javimangal@gmail.com

**Keywords:** liver steatosis, body fat distribution, nutrient intakes, saturated fatty acids, dietary sugars

## Abstract

**Background**: Metabolic dysfunction-associated steatotic liver disease (MASLD) is a prevalent disease influenced by dietary factors. While high sugar and fat consumption are associated with weight gain, their specific impact on liver fat accumulation remains unclear. This study aimed to evaluate the relationship between sugar and saturated fat intake and liver and body fat composition. **Methods**: A cross-sectional study was conducted from September 2021 to February 2023 in workers from a tertiary care center in Mexico City. Anthropometric measurements, body composition (bioelectrical impedance analysis and skinfold assessment), physical activity, and liver fat (vibration-controlled transient elastography) were measured. Dietary intake was assessed with a 24-h recall questionnaire and analyzed with specialized software. Linear and logistic regression models were fitted to study the relationship between nutrient intake and liver/body fat. **Results**: A total of 534 healthcare workers (median age: 41.5 years, 61.4% female) were included. Hepatic steatosis was present in 42.5% of participants. Higher carbohydrate intake was associated with increased liver fat (β = 0.23, 95% CI: 0.02–0.45), with each additional 15 g of carbohydrates increasing the odds of steatosis by 5% (OR = 1.053, 95% CI: 1.006–1.102). Fat and sugar intake were associated with higher body fat but not liver fat. **Conclusions**: Carbohydrate intake was linked to liver fat accumulation, whereas fat and sugar intake were primarily associated with body fat. Tailored dietary recommendations could be informed by these findings. Prospective dietary assessment methods and a nutritional geometry approach could be applied in future studies.

## 1. Introduction

Nonalcoholic fatty liver disease (NAFLD), now termed metabolic dysfunction-associated steatotic liver disease (MASLD), is a common global health problem affecting 25–35% of the population on average [1,2]. It has been described that diet and sedentary lifestyle influence the development of the disease, with diets high in sugars and fats being important risk factors [3,4,5,6,7,8,9,10,11,12].

The high consumption of energy-dense foods, especially sugars and fats, is associated with body weight gain. In a systematic review of longitudinal studies in adolescents and early adulthood, a diet pattern high in fast foods led to an excess odds of 23% (OR = 1.23; 1.02–1.49) of annual BMI gain of 0.08 kg/m^2^ [11]. Despite several efforts to determine the role of diet as a whole or dietary components as patterns, some relationships have only been partially explained, such as the consumption of fats and sugars in the diet [2]. Broadly, dietary lipids contribute ~15% to the pool of lipids deposited in the liver, whereas adipose tissue lipolysis and de novo lipogenesis account for 60–80% and 5%, respectively. Insulin regulates the use of adipose tissue sources of energy, which is an efficient process, while hepatocytes are involved in *de novo* lipogenesis from carbohydrates (especially fructose) under regulation of cytoplasmic transcription factors such as the farnesol X receptor from the PPAR’s family, of which the α and γ variants have anti-inflammatory activity, whereas the receptor suppresses lipogenesis and pro-inflammatory gene expression [13]. Lipid removal occurs through both mitochondrial oxidation and re-esterification to form triglycerides. These triglycerides, if stored as fat droplets, are released as fatty acids in hepatocytes when lipolyzed. In short, insulin resistance in the liver causes steatosis. Furthermore, it is known that excessive carbohydrate intake, particularly simple sugars (fructose and glucose), is linked to increased inflammatory factors and disease progression [2]. However, the quantity of each of these sugars associated with the development of fatty liver disease has not been established.

To date, American societies dedicated to the study of this disease have not issued a cut-off point for both dietary detection (risk factor) and lifestyle treatment regarding carbohydrates [14,15]. European associations did not have a clear recommendation in 2023 on the type and quantity of certain nutrients, including carbohydrates [16]. Nonetheless, it is important to note that a recent systematic review indicates that consumption of four or more servings of sugary drinks per week increases the risk of metabolic dysfunction-associated steatotic liver disease (MASLD) by 45% [17,18]. Regarding fat consumption, European, American, and Asian guidelines [14,17,19] align with the World Health Organization’s (WHO) recommendation to limit saturated fat intake to less than 10% of total energy [16]. To contextualize this, the most widely accepted hypothesis for the development of hepatic steatosis is the “multiple hit” model [13]. This model is characterized by:Steatosis due to excess triglycerides: Accumulation of triglycerides in the liver.Persistent insulin resistance: Leading to decreased glycogen synthesis, increased hepatic fatty acid uptake, altered triglyceride transport, and inhibited beta-oxidation.Pro-inflammatory cytokine activity: Driven by lipotoxicity, apoptosis, inflammasome activation, and mitochondrial dysfunction.Oxidative stress: An imbalance between pro-inflammatory and anti-inflammatory mechanisms that exacerbates insulin resistance in the presence of genetic or environmental susceptibility.Impaired hepatocyte regeneration and apoptosis.

This model explains the occurrence of hepatic steatosis in lean individuals, attributable to the predominance of genetic, metabolic, age-related, and epigenetic factors. In this population, greater heterogeneity in histological disease expression is observed. Therefore, screening from 40 years of age is generally recommended, and management should focus on a total weight loss not exceeding 5% [20]. To date, it remains inconclusive whether the type and quantity of sugars or fats influence fat deposition in specific body segments. Thus, the mechanisms by which body fat contributes to the development of metabolic liver disease are not established, although one study found that abdominal fat deposition is associated with greater metabolic imbalance [21].

The health status of healthcare workers has gained increasing relevance in recent years due to increasing rates of infections, cardiovascular diseases, malignant neoplasms, accidents, and injuries. Additionally, a significant prevalence of mental health issues, including stress and anxiety, has been observed, alongside a high frequency of overweight, respiratory diseases, and gastrointestinal disorders [22,23,24]. Despite the importance of this population group, there are limited reports evaluating hepatic steatosis and the knowledge of this disease within this context. A study conducted among healthcare workers in Brazil, evaluating the association between stress, anxiety, and steatosis, found a 26% prevalence of steatosis, with 15% of cases classified as mild and the remainder as moderate. However, no significant association was identified between the evaluated factors [25]. Regarding the knowledge of risk factors for this disease, a study in Mexico revealed that nearly 90% of healthcare workers consider hepatic steatosis to be a common condition, with metabolic syndrome, particularly obesity, being a relevant risk factor [26]. The scarcity of studies and preventive measures targeting this population is concerning, given their fundamental role in providing healthcare services. Therefore, the objective of this study was to study the relationship between sugar and saturated fat consumption and liver and body fat.

## 2. Methods

Analytical cross-sectional study conducted from September 2021 to February 2023 in the outpatient clinic of the Department of Gastroenterology, Salvador Zubirán National Institute of Medical Sciences and Nutrition, a tertiary care center in Mexico City.

### 2.1. Population

Institutional staff (medical, nurses, general clinical care, and administrative) were invited through an internal call. Eligible participants were adults over 18 years of age, of any sex, without a previous diagnosis of fatty liver disease. Participants were included if they self-reported no alcohol use disorder, hepatic steatosis, metabolic disorders under treatment, diabetes, or hypertension under treatment, and if they agreed to voluntarily participate in the interviews and study. Respondents were scheduled for biochemical and anthropometric measurements, questionnaire administration, and elastography at 7:00 A.M., coinciding with the shift change between night and morning duties. The average evaluation time was 2.5 h. All participants were informed and provided consent for participation in this study. Non-inclusion criteria were a diagnosis of cancer, heart disease, liver cirrhosis, hyperthyroidism, hypothyroidism, autoimmune diseases, bariatric or cosmetic surgery, metal prostheses, kidney failure, motor disability, amputations of limbs, use of medications that modify body composition (steroids, antipsychotics, antidepressants), and an alcohol consumption above 20 and 30 g per day for women and men, respectively [17]. Incomplete anthropometric assessments or vibration-controlled elastography were excluded from main analyses, but authorized 24-h reminders from all participants were included.

### 2.2. Demographic, Clinical, and Biochemical Data

A questionnaire was applied to obtain the demographic characteristics of the study population. Self-reported questions inquired about comorbidities (present/absent): diabetes mellitus, arterial hypertension, prior acute myocardial infarction, rheumatoid arthritis, dyslipidemia, hypothyroidism, insulin resistance, and tobacco use, as well as the self-reported job performed in the health institution, which was classified into medical, nursing, other clinical staff, administrative, and non-specified (when participants decided not to disclose their job task).

Anthropometric measurements were obtained with a SECA model 274 stadiometer (SECA, Hamburg, Germany) for height (precision ±2 mm) and a scale with bioelectrical impedance (mBCA514) for weight (±100 g). Body mass index (BMI) was calculated as the ratio of kg and squared meters (kg/m^2^) and categorized into universal BMI classes (<16.5, severely underweight; 16.5–18.5, underweight; 18.5–24.9, normal weight; 25–29.9, overweight; 30–34.9, class I obesity; 35–39.9, class II; and ≥40, class III obesity) [27]. All participants had blood samples taken for blood cytometry, blood chemistry, liver enzymes, lipid profile, C-reactive protein (CRP), and insulin to calculate the homeostatic model to assess insulin resistance (HOMA-IR); all laboratory parameters were obtained through the Beckman Coulter equipment (hematological DxH 1061 and series AU5800 for blood chemistry, Brea, CA, USA). We used the validated Laval questionnaire for the Mexican population to record physical activity for 24 h, which allows us to record the number of occasions that activity events occur in a day from fractions of 15 min and the type of activity. The result of each event is multiplied by a constant for the type of physical activity and divided by 60 min to obtain hours. These hours are multiplied by the person’s weight, and thus the energy expenditure for that activity is obtained. The total energy expenditure is obtained by adding all the individual expenses for each activity as Kcals per day [28]. Energy balance (Kcal) was calculated as the difference between the total energy expenditure by the Laval questionnaire and the energy consumption reported during the dietary evaluation. The results of total energy expenditure and energy balance were transformed into Z scores, and their standardized score is presented, as well as the classification of subjects < −1 SD, between −1 and 1 SD, and >1 SD.

### 2.3. Body Composition Assessment

Body composition measurements were made after the fasting period recommended for biochemical measurements. Multi-frequency bioelectrical impedance (BIA) analyses (11 frequencies) were performed using the SECA mBCA514 equipment. The data obtained from the BIA were total fat mass in kg and percentage and visceral fat in liters (L) [29,30].

All measurements were made with the anthropometric method validated by the International Society for the Advancement of Kinanthropometry by trained personnel who applied the Habitch technique. A Slim Guide caliper was used for body fold measurements (bicipital, tricipital, suprailiac, and subscapular skinfold), and a Lufkin metal tape model W606PM (Lufkin Industries, Missouri City, TX, USA) was used for arm and waist circumferences. Body fat percentage was estimated from the sum of skinfolds by applying the Durnin and Womersley formula [31].

### 2.4. Liver Fat Assessment

The evaluation of steatosis was performed consecutively with biochemical and body composition measurements to take advantage of the indicated 8-h fasting period. The degree of steatosis and liver fibrosis was assessed by vibration-controlled transient elastography (Fibroscan^®^ 502; Echosens, Paris, France) performed by trained physicians. The cut-off point used to determine steatosis was a controlled attenuation parameter (CAP) ≥ 275 dB/m [32] in accordance with European guidelines. Assessments that had 10 valid measurements and an IQR/med ≤ 30% were included.

### 2.5. Dietary Assessment

A 24-h multi-step reminder (Appendix A) was used for dietary assessment [33]. The analysis of the main nutrients, their types, and micronutrients was carried out with the Food Processor software v11.11^®^. The analysis included quantification in grams of total sugars, added sugars, fructose, and saturated fats, in addition to other types of nutrients. In total, 17 types of sugars, 5 types of fats, and total protein were quantified without distinction of their origin, estimated in terms of the amount of Kcal they represent (for each nutrient) and as a percentage of average total energy. The 31 micronutrients were expressed per day in the corresponding and universal dietary unit of measurement.

### 2.6. Sample Size

The sample size was calculated as the difference between two proportions, considering an overall prevalence of fatty liver of 25.2% [34] and an effect size of 11% associated with the consumption of sugary drinks on liver fat [11]. For this calculation, a confidence level of 95% and a statistical power of 80% were assumed. The minimum sample size needed to detect significant differences was 550 subjects. Calculations were performed using G*Power software, version 3.1.9.7.

### 2.7. Statistical Analysis

Descriptive data are presented as the median and interquartile range (Q1–Q3) for quantitative variables and frequency and percentage for qualitative variables. The comparisons in clinical, body composition, and laboratory data among participants with and without hepatic steatosis were made with the Mann–Whitney U test for quantitative variables and the Chi-square test or Fisher’s exact test for qualitative variables. The comparison of macronutrient and micronutrient intake was made using the Mann–Whitney U test.

To determine the relationship between nutrient intake and liver and body fat (CAP, body mass index, body fat, and waist circumference), different linear regression models were created for Kcal and each nutrient (carbohydrates, protein, fat, saturated fat, total sugars, added sugar, and fructose, each one in grams and in percentage of energy). For each model, the adjustment was made for age (quantitative), sex, BMI (quantitative), waist circumference (quantitative), and total Kcal (quantitative). The results were presented as the regression coefficient (β) with a 95% confidence interval (95%CI). The variance inflation factor (VIF) was calculated to determine the presence of collinearity in the multivariable models, defined as a value greater than 10.

The three evaluation methods recommended in nutritional epidemiology [35] were used to determine the degree of association between Kcal, nutrient intake, and hepatic steatosis (>275 dB/m):(1)Degree of association between nutrient intake (quantitative) and hepatic steatosis.(2)Nutrient intake distributed in quartiles, with quartile 1 as reference.(3)The association between the total sugar consumption > 10% (model 1) and saturated fat > 7% (model 2) with hepatic steatosis.

Both logistic regression models were adjusted for age (quantitative), sex, BMI (quantitative), waist circumference (quantitative), and total kilocalories (quantitative).

The results of the models are presented as the regression coefficient (β), standard error, odds ratio (OR), 95%CI of the OR, and *p*-value. The model assumptions were evaluated by residual analysis. The probabilities of developing fatty liver disease were calculated based on the regression coefficients from the multivariate models for each nutrient. The probability estimates were graphed with their 95% CIs for carbohydrate intake per gram and intake per 15 g.

A value of *p* < 0.05 was considered statistically significant. All analyses were performed using SPSS v21 software. Forest plots were created using GraphPad Prism v.9.1.1. Hepatic steatosis probabilities were obtained with R v. 4.4.3, with the “effects” package.

### 2.8. Ethical Procedures

Participants received and signed an informed consent form. In the event of not completing all study evaluations, authorization was requested to include all other completed assessments. This study was approved by the institutional research ethics committee with registration number GAS-3794. This study was conducted in compliance with the Declaration of Helsinki while maintaining the anonymity, privacy, and will of participants.

## 3. Results

A total of 583 eligible adults consented to participate in this study, of whom 49 were not included for analysis as elastography was not available. Of the 534 participants, 61.4% (n = 328) were female, and the median age was 41.5 (IQR: 29.0–52.0). The most frequently reported comorbidities were smoking (7.68%) and hypertension (3.98%). Furthermore, 30.7% (n = 164) of the participants had normal weight (BMI 18–24.9); 41.9% (n = 224) were overweight; 20.6% (n = 110) had class 1 obesity; 4.9% had class 2 obesity (n = 26); and 0.9% (n = 5) had class 3 obesity. A total of 493 participants had complete dietary analysis data. The flow of participants is depicted in Figure 1.

Table 1 presents demographic data, body composition, and results of the most relevant laboratory studies of people with and without hepatic steatosis. People with steatosis had older age and more metabolic alterations and anthropometric indices than those without steatosis. The type of work performed in healthcare was self-reported in four categories, and the response rate was only 46.3%, so most participants could not be classified according to their type of job. In both groups, the frequency of acute myocardial infarction and arthritis was low (≤1 case due to comorbidity per group). There were no differences between groups in urea nitrogen, urea, non-HDL cholesterol, liver function tests, or blood count.

The median CAP of the total sample was 263 dB/m (IQR: 211–304), while the median Kpa was 4.2 (IQR: 3.4–5.3). Of the 534 participants, 227 (42.5%, 95% CI: 38.3–46.7) had hepatic steatosis with a median CAP of 310 (IQR: 292–335) and Kpa of 4.6 (IQR: 3.7–5.8). Of the 307 (57.5%) who did not have steatosis, the median CAP was 193 dB/m (IQR: 221–250) with Kpa of 3.3 (IQR: 4.0–4.8).

Table 2 presents the results of the comparisons of nutrient and micronutrient intake between subjects with and without hepatic steatosis. Intake of energy, protein, total fat, and saturated fat was higher in people without hepatic steatosis. Although carbohydrate consumption was similar between groups, monosaccharide consumption, particularly fructose, was higher in subjects without steatosis. An adjustment was made between the main nutrients by the kilocalories of consumption, thus obtaining the percentage of energy consumed in the form of each nutrient. It was observed that after this adjustment, there were no differences in consumption between the study groups. Regarding micronutrients, no differences were observed between those with and without hepatic steatosis.

Table 3 presents the results of multivariable linear regression models to determine the relationship between nutrient intake and the amount of liver fat, determined by CAP, body fat, visceral fat, and waist circumference. As the percentage of carbohydrate consumption increased, the CAP increased, and therefore the amount of liver fat (β = 0.23, 95%CI: 0.02 to 0.45). Likewise, the increase in the percentage of fat consumption was related to lower values of CAP (β = −0.22, 95%CI: −0.44 to −0.006). The consumption of Kcal, total fat, total sugars, added sugar, and percentage of fat consumption is related to the increase in the percentage of body fat, while lower intakes of total carbohydrates (g), total protein (g), and percentages of carbohydrates were related to increasing body fat. Waist circumference had a weak relationship with Kcal intake, and visceral fat was not associated with any nutrient.

Figure 2 presents the results of logistic regression models for determining the association of nutrient intake with hepatic steatosis. It was observed that carbohydrate intake was associated with higher odds of fatty liver. Each gram of consumption was associated with a 0.3% increase in the odds of having fatty liver (β = 0.003, *p* = 0.03). We performed a regression model adjusting carbohydrate consumption per 15 g, showing that the odds of having fatty liver increased by 5% (β = 0.51, OR = 1.053, 95%CI: 1.006–1.102, *p* = 0.03) for every 15 g. In Figure 3 we present the impact of carbohydrate intake on the probability of having hepatic steatosis. It can be observed how the probability of hepatic steatosis increased as carbohydrate consumption increased by 1 g or 15 g.

Appendix A presents the results of logistic regression models, considering the distributions of nutritional intake by quartiles. No association was observed between higher intake and increased risk of hepatic steatosis. For the purposes of this analysis, participants were categorized based on adherence to dietary recommendations, specifically those consuming free sugars at less than 10% of total energy intake [20] and saturated fat at less than 7% of total energy intake [36]. Consumption of sugars > 10% (n = 217, 44%, OR = 1.04, 95%CI: 0.67–1.60, *p* = 0.86) or saturated fats > 7% (n = 451, 91.5%, OR = 0.92, 95%CI: 0.43–1.96, *p* = 0.83) was not associated with the presence of hepatic steatosis in quartile analyses.

Finally, the frequency of subjects with combined intakes of fats, total sugars, added sugars, and fructose was explored by considering those within quartile 4 of each of the nutrients as high consumption. The frequency of combined high consumption was low for all combinations: fats with total sugars (n = 7, 1.3%), fats with added sugars (n = 6, 1.1%), and fat with fructose (n = 20, 3.7%).

## 4. Discussion

The objective of this study was to analyze the association between nutrient, sugar, and saturated fat consumption with liver and body fat in the general population. Our results showed that only carbohydrate intake was related to liver fat, with each 15-g serving increasing the odds of developing hepatic steatosis by 5%. Furthermore, higher energy intake from fats and sugars was associated with increased body fat but had no significant effect on waist circumference or visceral fat. These findings provide insight into how diet influences fat accumulation but do not establish its impact on central fat distribution.

In this study, we used information on self-reported comorbidities. Few reports describe the frequency of self-recognition of the disease in general [37] and in-hospital environments [38]. Therefore, these are insightful reports, although future studies could seek to make rigorous confirmation of comorbidities. Participants diagnosed with steatosis were observed to be older and exhibited a higher incidence of metabolic abnormalities, including elevated HOMA, glucose, LDL cholesterol, triglycerides, and insulin, coupled with decreased HDL cholesterol. Anthropometric measurements, particularly waist circumference exceeding 90 cm, demonstrated a strong correlation with MASLD, consistent with previous reports [39,40]. Eight percent had normal weight and steatosis, as reported in the literature [20]. We observed that the probability of presenting hepatic steatosis increases with carbohydrate consumption. Furthermore, we observed an increased probability of body fat accumulation associated with a diet high in fats and sugars, consistent with previous reports [2].

The WHO [36] recommends that a regular diet should contain less than 10% free sugars and minimal saturated fats. Similarly, the National Cholesterol Education Program [41] suggests that a non-atherogenic diet should contain less than 7% saturated fat. However, in this cross-sectional study, we found no association between hepatic steatosis and consumption exceeding these values, which is why we considered 15 g of carbohydrates to be an increase in relevance, which is the recommended portion for carbohydrates in diabetes [42].

Our study revealed an apparent discrepancy between the presence of steatosis and dietary intake, as patients with steatosis reported lower energy, fat, and monosaccharide intake. Several factors could explain this finding. First, patients with steatotic liver disease, who had a higher BMI, may have underreported their intake. Weight bias has been shown to lead to underreporting of calorie intake by nearly 500 kcal in obese individuals [43], and this effect may be even more pronounced among healthcare professionals, who comprised our study population. Considering that we also observed underreporting in the physical activity questionnaire, it was striking that the average energy balance in our study was positive by approximately 600 kcals. One possible explanation for this could be reverse causation if metabolic conditions such as overweight and obesity had prompted dietary modifications. Given that our participants were healthcare personnel, their heightened awareness of health risks may have influenced their dietary behaviors.

Regarding fat intake, our study found an association between higher total fat intake and reduced fatty liver, a finding that remains unclear. Previous studies, including a randomized controlled trial comparing the effects of saturated versus polyunsaturated fatty acids on liver fat, have confirmed that excessive saturated fat intake promotes liver steatosis [44]. However, we could not replicate this association, as saturated fat intake was similar between groups. One possible explanation for these inconsistencies is that the food source of fats may play a more significant role than the nutrient itself. For instance, extra-virgin olive oil, a major source of polyunsaturated fatty acids, has been consistently shown to have a protective effect against MASLD, even in RCTs [45,46]. Therefore, it is essential to develop studies that evaluate food sources, whether in food groups or through dietary patterns.

Similarly, we did not find an association between fructose intake and hepatic steatosis. This may be because the metabolic effects of fructose depend on its food source rather than the nutrient itself. For example, a systematic review and meta-analysis of 51 trials found that the adverse effects of fructose were most pronounced when derived from sugar-sweetened beverages [47].

Our findings suggest that rather than focusing solely on macronutrient composition, other dietary aspects should be considered when assessing the role of nutrition in MASLD. Some authors propose that a diet’s inflammatory potential may be more relevant, as pro-inflammatory diets have been associated with MASLD diagnosis and progression [48]. Consumption patterns may also play a crucial role. For example, intermittent fasting has been suggested as a strategy to reverse lipid accumulation in the blood and liver, particularly in high-fat diets [13]. A recent RCT comparing intermittent fasting with calorie restriction found that time-restricted eating provided additional benefits for MASLD, independent of weight loss [49]. Similarly, dietary patterns seem to be very important, and the Mediterranean diet is widely recommended, with a recent systematic review and meta-analysis of 15 studies supporting its efficacy in reducing intrahepatic lipid content compared to a general low-calorie diet [50].

Other sources of inconsistency may stem from failing to account for factors such as nutrient bioavailability and origin. Additionally, genetics, microbiota composition, and epigenetics likely modulate these associations [2]. In particular, nutrigenetics, the study of how genetic variation affects dietary response, may impact the understanding of the role of specific nutrients on liver health [51]. These complexities highlight the challenge of interpreting the effects of isolated nutrients, as dietary components interact in multifaceted ways. Future studies should consider a nutritional geometry approach, which evaluates the interplay between nutrients, foods, appetite regulation, and metabolic homeostasis [52]. However, even this approach has not yet established an optimal balance of nutrients to support metabolic health.

Our study has limitations that may explain some of the discrepancies between our results and previous findings. The use of a single 24-h dietary recall (R24) to assess intake may have been inaccurate. More precise dietary assessment could be achieved through repeated R24 measurements, and emerging digital tools have been proposed to improve accuracy using automated image-based food recognition [53]. The current recommendation for accurate dietary estimation is the application of four R24 [54]. Due to budgetary limitations, we were restricted to the application of a single questionnaire. The lower caloric intake reported by participants with hepatic steatosis despite their higher metabolic risk suggests that the dietary assessment may have been subject to reporting bias or confounding by indication, in which individuals aware of their metabolic risk may have modified their diet. Similarly, a relationship has been observed between increasing BMI and under-reporting of fat and energy consumption in several assessment modalities, including R24 [55]. This seems to have occurred in our study, as there appears to be under-reporting in subjects who would have a greater metabolic alteration. Lastly, we performed multiple statistical comparisons without *p*-value adjustment, reason why the results should be interpreted as suggestive, and possibly false positive findings.

The generalizability of our findings may be limited to populations with characteristics similar to healthcare workers in Mexico, who tend to be young, have low physical activity levels, consume low fiber and excessive processed meats and sweetened beverages, and are undergoing a dietary transition toward higher fat, carbohydrate, and sodium intake [56]. Furthermore, the multiple exclusion criteria in our study may restrict the applicability of the findings in other populations. Another limitation was our inability to assess dietary patterns over time or determine the intake of specific dietary components such as omega-3 and omega-6 fatty acids, polyphenols, or carotenoids, which have been recognized as metabolic protective factors [2]. Additionally, we did not collect information on specific food sources. Hepatic steatosis was diagnosed using CAP measurements, which, while more sensitive than ultrasound, may not reliably detect mild steatosis. Moreover, CAP cut-off values vary in the literature, adding another layer of complexity to interpretation. We believe that awareness of self-perception and health recognition, as well as the phenomenology of illness, are terms that should be integrated into the workplace and the education of the healthcare worker community. This is because, in addition to the scarcity of evidence, there is also a significant gap in health systems, and it is not until irreversible outcomes such as disability or mortality occur that those who suffer are taken into account [37,38]. Finally, another limitation of our study was the assessment of alcohol consumption solely through self-reporting. Recent evidence shows that in some cohorts initially classified as pure MASLD, the use of serum alcohol biomarkers reduces the proportion of pure MASLD cases and increases the number of patients reclassified into the MetALD (metabolic and alcohol-related liver disease) group [57].

## 5. Conclusions

In conclusion, our study found that carbohydrate intake was associated with liver fat accumulation, whereas fat and sugar intake were primarily linked to body fat. Future research should incorporate prospective dietary assessments and a nutritional geometry approach to better understand these relationships. Considering factors such as food sources, dietary patterns, and consumption habits will be essential in elucidating the role of diet in MASLD.

## Figures and Tables

**Figure 1 nutrients-17-01328-f001:**
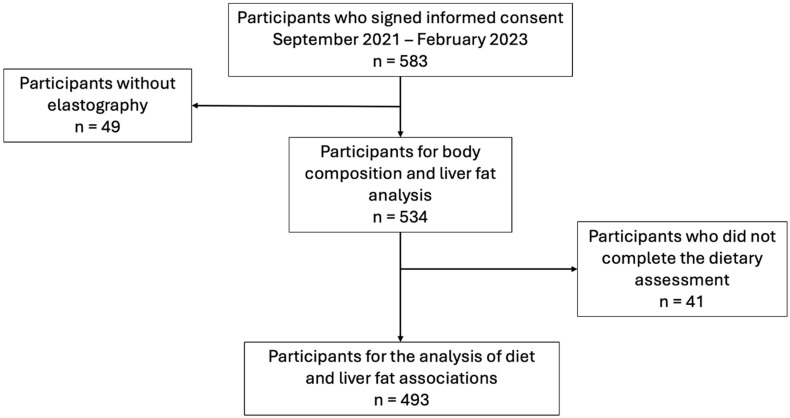
Flowchart of the participants.

**Figure 2 nutrients-17-01328-f002:**
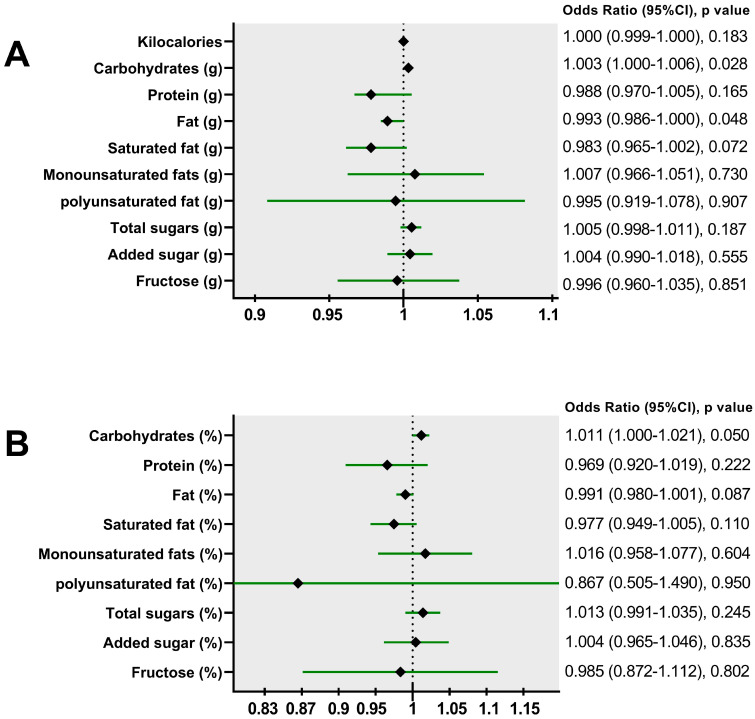
Multivariable logistic regression models of the relationship between dietary intake and the odds of hepatic steatosis. (**A**) Nutrients in grams. (**B**) Nutrients in percentage of total energy. Models adjusted for age, sex, BMI, waist circumference, and total Kcal. Kilocalorie models adjusted for age, sex, BMI, and waist circumference. Black diamond represent the Odds Ratio, and the green line the 95%CI.

**Figure 3 nutrients-17-01328-f003:**
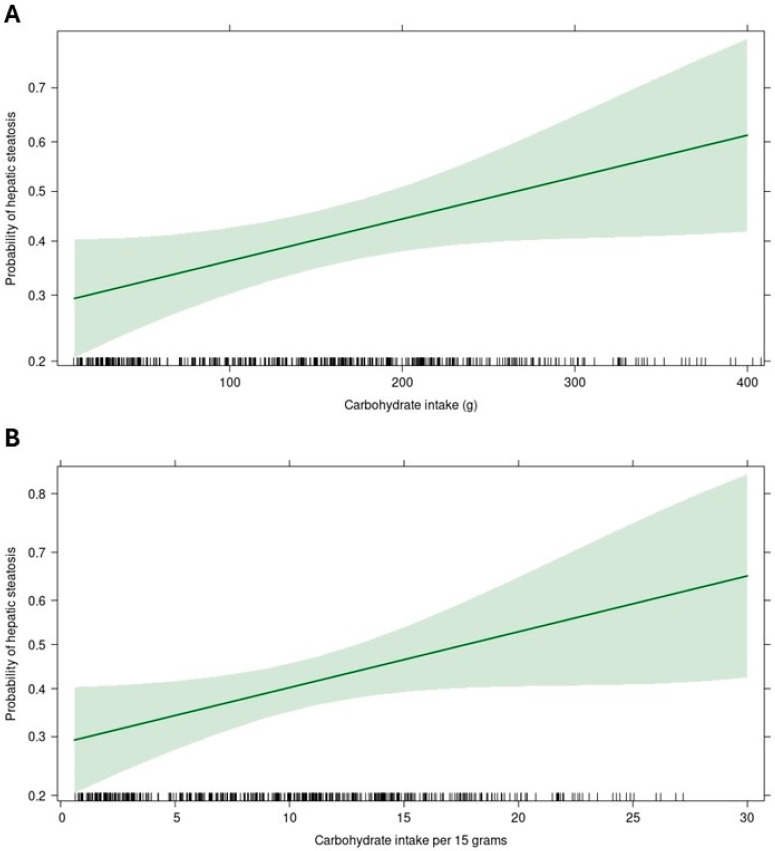
Impact of carbohydrate intake on the likelihood of developing fatty liver disease. (**A**) Carbohydrate intake in grams. (**B**) Carbohydrate intake per 15 g. Probabilities adjusted for age, sex, BMI, waist circumference, and total Kcal. The probability estimate (green line) and its 95% confidence interval (green highlight) are shown.

**Table 1 nutrients-17-01328-t001:** Clinical, body composition, and laboratory data among participants with and without hepatic steatosis.

	Total Sample (n = 534)	No Steatosis (n = 307)	With Steatosis (n = 227)	*p* Value
Age (years)	41.5 (29.0–52.0)	36.0 (27.0–51.0)	45.0 (34.0–53.0)	<0.0001
**Sex, n (%)**				
Women	328 (61.4)	199 (64.8)	129 (56.8)	0.061
**Smoking, n (%)**	41 (7.68)	18 (5.86)	23 (10.13)	0.062
**Comorbidities, n (%)**				
Diabetes	9 (1.69)	3 (0.98)	6 (2.64)	0.178 *
Hypertension	21 (3.93)	9 (2.93)	12 (5.29)	0.166
Dyslipidemia	3 (0.56)	0 (0.00)	3 (1.32)	0.077
Hypothyroidism	9 (1.69)	5 (1.63)	4 (1.76)	0.999
Insulin resistance	4 (0.75)	1 (0.33)	3 (1.32)	0.316
**Job categories**				
Medical staff	31 (5.8)	20 (6.5)	11 (4.8)	0.857
Nursing staff	46 (8.6)	23 (7.5)	23 (10.1)	
Non-medical clinical staff	43 (8.1)	26 (8.5)	17 (7.5)	
Administrative staff	126 (23.8)	74 (24.1)	52 (22.9)	
Not disclosed	288 (53.9)	164 (53.4)	124 (54.6)	
**Body composition**				
Weight (kg)	70.5 (60.6–80.7)	64.2 (56.7–74.5)	77.8 (69.4–86.4)	<0.0001
BMI	27.1 (23.8–30.3)	25.1 (22.3–27.8)	29.4 (27.1–32.8)	<0.0001
<18.5	5 (0.9)	5 (1.6)	0 (0.0)	<0.0001
18–24.9	164 (30.7)	146 (47.6)	18 (7.9)	
25–29.9	224 (41.9)	117 (38.1)	107 (47.1)	
30–34.9	110 (20.6)	34 (11.1)	76 (33.5)	
35–39.9	26 (4.9)	3 (1.0)	23 (10.1)	
≥40	5 (0.09)	2 (0.7)	3 (1.3)	
Fat mass (kg)	25.1 (18.7–31.5)	20.8 (15.5–27.4)	29.6 (25.2–35.8)	<0.0001
Fat mass (%)	35.6 (29.0–42.2)	32.9 (26.2–38.7)	39.7 (32.6–44.7)	<0.0001
Visceral fat (L)	2.50 (1.80–3.50)	2.00 (1.50–2.70)	3.20 (2.48–4.20)	<0.0001
Waist circumference (m)	0.89 (0.80–0.99)	0.84 (0.76–0.92)	0.96 (0.89–1.04)	<0.0001
Bicipital skinfold (mm)	10.0 (7.0–14.0)	8.0 (6.0–12.0)	12.0 (9.0–17.0)	<0.0001
Triceps skinfold (mm)	17.0 (12.0–22.0)	15.0 (12.0–19.0)	20.0 (14.0–25.0)	<0.0001
Subscapular skinfold (mm)	23.0 (17.0–30.0)	19.0 (15.0–25.0)	28.0 (23.0–33.0)	<0.0001
Suprailiac skinfold (mm)	24.0 (17.0–30.0)	20.0 (15.0–27.0)	28.0 (24.0–35.0)	<0.0001
**Physical activity**				
Kcals from PAQ *	737.7 (636.5–852.2)	699.4 (600.9–814.2)	781.3 (697.7–928.2)	<0.0001
Z Score PAQ *	0.00 (1.00)	−0.27 (0.89)	0.36 (−0.13)	<0.0001
<−1 SD	75 (14.0)	65 (22.0)	10 (4.5)	<0.0001
−1 to 1 SD	368 (68.9)	203 (68.6)	165 (74.3)	
>1 SD	75 (14.0)	28 (9.5)	47 (21.2)	
Energy balance	529.5 (203.1–920.9)	557.9 (249.8–980.9)	463.00 (153.1–829.4)	0.013
Z Score Energy balance	0.00 (1.00)	0.09 (1.01)	−0.13 (0.97)	0.01
<−1 SD	84 (15.7)	14 (13.0)	44 (19.4)	0.04
−1 to 1 SD	370 (69.3)	216 (70.4)	154 (67.8)	
>1 SD	80 (15.0)	51 (16.6)	29 (12.8)	
**Biochemical data**				
HOMA IR	1.58 (1.04–2.62)	1.25 (0.81–1.78)	2.45 (1.55–3.85)	<0.0001
Glucose (mg/dL)	89.0 (84.0–96.0)	87.0 (82.0–92.0)	93.0 (88.0–101.0)	<0.0001
Creatinine (mg/dL)	0.76 (0.66–0.89)	0.74 (0.66–0.87)	0.77 (0.66–0.92)	0.152
Cholesterol (mg/dL)	180.0 (155.0–206.0)	177.0 (155.0–202.0)	185.0 (156.0–217.0)	0.009
Low-density cholesterol (LDL-c; mg/dL)	110.0 (89.0–129.0)	105.0 (86.0–124.0)	116.0 (92.5–134.5)	0.001
LDL-c, Martin’s method (mg/dL)	107.0 (86.0–126.0)	104.0 (83.0–122.0)	112.0 (88.0–133.0)	0.003
High-density cholesterol (HDL-c; mg/dL)	47.0 (40.0–56.0)	50.0 (43.0–59.0)	43.0 (37.0–51.0)	<0.0001
Triglycerides (mg/dL)	124.0 (90.0–174.0)	106.0 (79.0–142.0)	163.0 (114.5–233.0)	<0.0001
Total bilirubin (mg/dL)	0.64 (0.49–0.85)	0.64 (0.49–0.85)	0.63 (0.51–0.83)	0.762
Alanine aminotransferase (ALT; U/L)	21.6 (15.4–32.2)	18.1 (13.9–26.0)	26.5 (18.0–40.7)	<0.0001
Aspartate aminotransferase (AST; U/L)	19.6 (17.0–24.8)	19.1 (16.9–23.7)	21.4 (17.2–26.7)	0.002
Gammaglutamyl transferase (U/L)	21.4 (15.1–35.3)	17.9 (13.1–26.7)	27.9 (19.1–42.6)	<0.0001
Alkaline phosphatase (U/L)	74.0 (61.5–88.0)	71.0 (58.0–85.0)	78.0 (66.0–92.0)	<0.0001
Albumin (g/dL)	4.42 (4.23–4.62)	4.46 (4.25–4.64)	4.38 (4.22–4.59)	0.066
Ultra-sensitive C-reactive protein (mg/dL)	0.15 (0.08–0.31)	0.13 (0.06–0.24)	0.20 (0.11–0.41)	<0.0001
Insulin (μIU/mL)	7.19 (4.95–11.38)	5.72 (3.96–8.15)	10.43 (6.91–16.02)	<0.0001
Platelets (10^3^/μL)	249.0 (213.0–289.5)	251.0 (214.0–290.0)	242.5 (211.0–287.2)	0.496

Data are presented as frequency and percentage (%) or as median and interquartile range (Q1–Q3). Quantitative variables were compared with the Mann–Whitney U test, whereas qualitative variables were compared using chi-square test or Fisher’s exact test. * The Laval physical activity questionnaire [28].

**Table 2 nutrients-17-01328-t002:** Comparison of energy and nutrient intake between participants with and without fatty liver collected with 24-h recall.

	Total Sample (n = 493)	No Steatosis (n = 282)	With Steatosis (n = 211)	*p* Value
Kilocalories	1278.6 (1021.5–1533.9)	1326.2 (1050.7–1557.7)	1188.7 (991.8–1499.0)	0.009
**Nutrients**				
Carbohydrates (g)	143.6 (67.5–211.7)	145.1 (53.6–216.96)	142.9 (78.0–208.5)	0.959
Protein (g)	55.2 (44.7–69.0)	56.4 (46.5–70.5)	53.1 (40.8–65.8)	0.019
Fat (g)	43.1 (30.6–75.2)	45.2 (32.5–86.6)	41.3 (28.8–62.3)	0.008
Kilocalories per carbohydrate	574.5 (270.1–846.7)	580.2 (214.4–867.8)	571.7 (312.2–834.1)	0.959
Carbohydrates (%)	53.7 (28.2–60.0)	51.8 (13.6–60.0)	55.4 (41.2–60.0)	0.253
Kilocalories from protein	221.0 (179.0–276.2)	225.9 (186.2–281.9)	212.3 (163.5–263.4)	0.019
Protein (%)	16.6 (15.2–18.9)	16.9 (15.4–19.4)	16.5 (15.0–18.6)	0.241
Kilocalories from fat	387.7 (275.7–676.8)	407.3 (293.1–780.1)	372.0 (259.2–560.9)	0.008
Fats (%)	26.8 (23.2–49.6)	27.8 (23.2–70.5)	26.8 (23.1–39.4)	0.435
Saturated fat (g)	16.8 (11.6–28.6)	17.7 (12.1–32.1)	15.8 (10.9–24.9)	0.022
Kilocalories per saturated fat	151.4 (104.3–257.9)	159.4 (109.1–288.7)	142.4 (98.4–224.9)	0.022
Saturated Fat (%)	10.6 (9.05–18.35)	10.6 (9.11–25.98)	10.5 (9.05–14.82)	0.587
Monounsaturated fat (g)	3.91 (1.56–7.47)	3.98 (1.62–7.56)	3.50 (1.41–7.24)	0.516
Polyunsaturated fat (g)	1.76 (0.81–3.36)	2.02 (0.90–3.39)	1.60 (0.78–3.34)	0.232
Trans fats (g)	0.00 (0.00–0.12)	0.00 (0.00–0.08)	0.00 (0.00–0.13)	0.250
**Types of carbohydrates**				
Available CH (g)	134.2 (57.8–197.5)	134.3 (47.5–202.4)	134.2 (68.4–190.0)	0.886
Total sugars (g)	28.0 (17.98–55.39)	28.4 (18.61–52.03)	28.0 (16.68–55.99)	0.352
Added sugar (g)	7.53 (3.39–20.14)	7.53 (3.52–21.04)	7.67 (3.28–19.36)	0.649
Total sugars (%)	8.57 (6.43–14.73)	8.57 (6.43–14.64)	8.57 (6.43–16.14)	0.728
Added sugars (%)	2.44 (1.30–5.65)	2.43 (1.30–5.71)	2.49 (1.25–5.63)	0.845
Monosaccharides (g)	7.80 (3.69–12.40)	8.44 (4.40–12.72)	6.76 (2.33–12.00)	0.037
Galactose (g)	0.00 (0.00–0.00)	0.00 (0.00–0.00)	0.00 (0.00–0.00)	0.794
Glucose (g)	2.91 (1.52–4.71)	3.06 (1.99–4.67)	2.51 (0.99–4.72)	0.049
Fructose (g)	4.86 (1.97–8.78)	5.42 (2.28–8.78)	4.35 (0.96–7.69)	0.017
Fructose (%)	1.56 (0.75–2.45)	1.65 (0.77–2.46)	1.50 (0.42–2.45)	0.266
Disaccharides (g)	2.28 (1.18–3.75)	2.37 (1.41–3.75)	2.11 (0.71–3.62)	0.127
Lactose (g)	0.00 (0.00–0.00)	0.00 (0.00–0.00)	0.00 (0.00–0.00)	0.834
Maltose (g)	0.00 (0.00–0.01)	0.00 (0.00–0.01)	0.00 (0.00–0.01)	0.404
Other HC (g)	75.8 (15.3–142.7)	68.7 (15.1–143.3)	78.9 (15.6–138.6)	0.871
Net HC (g)	134.4 (63.1–197.5)	134.4 (47.9–202.0)	134.4 (68.4–190.0)	0.938
Non-digestible HC (g)	0.00 (0.00–0.00)	0.00 (0.00–0.00)	0.00 (0.00–0.00)	0.833
Dietary fiber (g)	8.57 (4.96–13.36)	8.56 (4.97–13.72)	8.57 (4.95–13.16)	0.807
Starch (g)	0.00 (0.00–1.84)	0.00 (0.00–1.84)	0.00 (0.00–1.83)	0.418
**Micronutrients**				
Cholesterol (mg)	132.5 (83.4–217.4)	141.7 (94.1–221.1)	130.2 (78.8–205.9)	0.063
Folate (μg)	106.9 (58.7–157.1)	108.4 (59.4–159.3)	104.9 (54.9–156.9)	0.458
Folic acid (μg)	13.9 (0.00–62.3)	7.90 (0.00–66.1)	14.28 (0.00–60.1)	0.478
Vitamin B1 (mg)	0.48 (0.25–0.71)	0.49 (0.27–0.77)	0.45 (0.24–0.70)	0.158
Vitamin B2 (mg)	0.88 (0.58–1.28)	0.92 (0.58–1.33)	0.84 (0.57–1.26)	0.519
Vitamin B3 (mg)	12.0 (8.26–19.04)	12.4 (8.46–19.43)	11.6 (8.17–18.67)	0.468
Pantothenic acid (mg)	0.57 (0.25–1.02)	0.57 (0.19–1.02)	0.57 (0.27–1.05)	0.505
Vitamin B6 (mg)	0.85 (0.50–1.34)	0.92 (0.50–1.40)	0.79 (0.50–1.27)	0.158
Vitamin B12 (μg)	1.12 (0.41–2.37)	1.12 (0.37–2.45)	1.07 (0.41–2.35)	0.746
Vitamin C (mg)	42.3 (13.7–96.4)	40.1 (14.6–96.9)	42.9 (11.6–96.4)	0.952
Vitamin D (μg)	0.18 (0.00–0.79)	0.19 (0.00–0.74)	0.18 (0.00–0.82)	0.490
Vitamin E (mg)	0.48 (0.23–1.02)	0.50 (0.23–1.11)	0.47 (0.22–0.95)	0.521
Vitamin K (μg)	14.7 (5.03–51.57)	14.6 (5.19–52.50)	15.0 (4.62–49.98)	0.483
Biotin (μg)	3.59 (0.50–7.27)	3.68 (1.33–7.27)	3.49 (0.36–7.27)	0.401
Vitamin A (IU)	426.9 (135.4–856.3)	413.5 (124.9–863.7)	440.1 (155.9–849.0)	0.691
Calcium (mg)	916.6 (517.7–1311.1)	875.1 (453.7–1299.9)	958.4 (572.5–1320.7)	0.152
Copper (mg)	0.34 (0.21–0.53)	0.35 (0.21–0.54)	0.32 (0.21–0.51)	0.517
Iron (mg)	12.68 (7.52–18.20)	12.89 (7.35–18.56)	12.56 (8.04–17.71)	0.929
Magnesium (mg)	109.5 (62.3–163.3)	114.2 (62.6–173.3)	104.2 (61.6–157.5)	0.574
Boron (μg)	155.1 (0.00–529.19)	248.4 (0.00–529.19)	122.1 (0.00–529.19)	0.102
Chlorine (mg)	62.0 (0.0–150.0)	84.7 (0.0–150.0)	46.8 (0.0–150.0)	0.193
Chromium (μg)	1.39 (0.00–2.32)	1.39 (0.00–2.54)	1.05 (0.00–2.32)	0.380
Fluoride (mg)	0.00 (0.00–0.01)	0.00 (0.00–0.01)	0.00 (0.00–0.01)	0.841
Iodine (μg)	1.13 (0.15–33.51)	1.42 (0.15–36.74)	0.88 (0.12–19.79)	0.147
Manganese (mg)	0.32 (0.10–0.58)	0.32 (0.09–0.58)	0.32 (0.10–0.58)	0.657
Molybdenum (μg)	0.00 (0.00–1.26)	0.00 (0.00–1.29)	0.00 (0.00–1.22)	0.833
Phosphorus (mg)	490.8 (305.7–724.6)	499.5 (300.4–772.5)	486.1 (309.1–684.1)	0.669
Potassium (mg)	1404.2 (845.2–1957.7)	1424.5 (807.9–1986.6)	1373.7 (888.0–1953.6)	0.833
Selenium (μg)	45.58 (30.59–70.16)	46.53 (30.36–72.71)	44.96 (30.59–68.29)	0.562
Sodium (mg)	2332.6 (1597.9–3033.6)	2335.7 (1589.2–3146.7)	2289.8 (1610.1–2968.2)	0.578
Zinc (mg)	3.73 (1.76–8.89)	4.33 (1.66–9.27)	3.47 (1.87–8.22)	0.496

Data presented as median and interquartile range (Q3–Q1). Comparisons were made using the Mann–Whitney U test. HC: carbohydrates.

**Table 3 nutrients-17-01328-t003:** Multivariable linear regression models of the relationship between dietary intake and liver and body fat.

	CAP Model		Fat Model		Model Waist		Visceral Fat Model	
Nutrient	B (95% CI)	*p*-Value	B (95% CI)	*p*-Value	B (95% CI)	*p*-Value	B (95% CI)	*p*-Value
Kilocalories ‡	−0.006 (−0.016 to 0.004)	0.204	0.001 (0.044–0.000)	0.002	−0.001 (−0.002–0.000)	0.042	0.000 (0.000–0.000)	0.567
Carbohydrates (g) *	0.060 (−0.001 to 0.121)	0.053	−0.002 (0.372–−0.007)	0.003	0.001 (−0.006–0.007)	0.862	0.000 (0.000–0.001)	0.589
Protein (g) *	−0.082 (−0.434 to 0.271)	0.649	−0.009 (0.541–−0.039)	0.020	0.002 (−0.034–0.038)	0.900	−0.001 (−0.005–0.003)	0.682
Fat (g) *	−0.129 (−0.266 to 0.007)	0.063	0.006 (0.318–−0.006)	0.017	−0.001 (−0.015–0.013)	0.845	0.000 (−0.002–0.001)	0.639
Saturated fat (g) *	−0.325 (−0.706–0.056)	0.094	0.015 (0.347–−0.017)	0.047	−0.006 (−0.045–0.033)	0.747	−0.001 (−0.005–0.003)	0.701
Monounsaturated Fats (g)	−0.037 (−0.952–0.878)	0.937	0.017 (−0.058–0.092)	0.658	0.047 (−0.045–0.138)	0.316	−0.007 (−0.017–0.003)	0.149
Polyunsaturated Fats (g)	0.358 (−1.339–2.055)	0.678	0.002 (−0.137–0.142)	0.975	0.083 (−0.087–0.253)	0.337	−0.006 (−0.024–0.012)	0.502
Total sugars (g) *	0.054 (−0.087–0.195)	0.453	0.005 (0.395–−0.007)	0.017	−0.009 (−0.023–0.006)	0.237	0.000 (−0.001–0.002)	0.812
Added sugar (g) *	0.098 (−0.190–0.386)	0.506	0.018 (0.148–−0.006)	0.042	−0.012 (−0.041–0.017)	0.425	0.000 (−0.003–0.003)	0.834
Fructose (g) *	−0.129 (−0.920–0.661)	0.748	0.026 (0.448–−0.041)	0.092	−0.052 (−0.132–0.029)	0.210	−0.001 (−0.009–0.008)	0.884
Protein (%) *	−0.285 (−1.336–0.766)	0.594	−0.019 (0.679–−0.106)	0.069	0.006 (−0.101–0.113)	0.912	−0.001 (−0.012–0.010)	0.862
Carbohydrates (%) *	0.234 (0.019–0.449)	0.033	−0.013 (0.155–−0.031)	0.005	0.000 (−0.022–0.022)	0.977	0.001 (−0.001–0.004)	0.247
Fat (%) *	−0.220 (−0.435–0.006)	0.044	0.014 (0.133–−0.004)	0.032	0.000 (−0.022–0.022)	0.995	−0.001 (−0.04–0.001)	0.263
Saturated Fat (%) *	−0.557 (−1.145–0.031)	0.063	0.037 (0.145–−0.013)	0.086	−0.003 (−0.064–0.057)	0.910	−0.003 (−0.010–0.003)	0.329
Monounsaturated Fats (%)	0.163 (−1.113–1.438)	0.802	0.012 (−0.093–0.117)	0.818	0.085 (−0.043–0.212)	0.193	−0.009 (−0.023–0.005)	0.201
Polyunsaturated Fats (%)	0.819 (−1.593–3.232)	0.505	−0.020 (−0.219–0.178)	0.841	0.127 (−0.115–0.369)	0.302	−0.007 (−0.033–0.019)	0.582
Total sugars (%) *	0.165 (−0.293–0.622)	0.480	0.009 (0.651–−0.029)	0.047	−0.028 (−0.075–0.018)	0.236	0.000 (−0.005–0.005)	0.954
Added sugar (%) *	0.211 (−0.649–1.072)	0.630	0.034 (0.355–−0.038)	0.106	−0.030 (−0.118–0.058)	0.503	−0.001 (−0.011–0.008)	0.791
Fructose (%) *	−0.719 (−3.298–1.859)	0.584	0.049 (0.654–−0.167)	0.266	−0.158 (−0.422–0.106)	0.240	−0.005 (−0.033–0.023)	0.725

‡: Models adjusted for age, sex, BMI, and waist circumference. *: Models adjusted for age, sex, BMI, waist circumference, and total Kcal.

## Data Availability

The data presented in this study are available upon request from the corresponding author. The data are not publicly available because they contain information that could compromise the privacy of research participants.

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
