# Peer review of "The Impact of Dietary Sugars and Saturated Fats on Body and Liver Fat in a Healthcare Worker Population"

_nutrients, 2025, doi:10.3390/nu17081328_

Round 1
Reviewer 1 Report
Comments and Suggestions for Authors
Dear Authors,
Thank you for submitting your manuscript. Please see my comments below.
The Introduction section is quite brief. Additional information on the physiological mechanisms underlying the effects of diet on fatty liver should be provided. Furthermore, evidence suggests that high sugar consumption predicts fatty liver, and this should be included. Additionally, it is necessary to clearly outline the novelty and significance of conducting this study specifically among healthcare workers. Why is this particular group important?
Please ensure references are provided to support all study measures, such as bioelectric impedance and dietary assessments. Additionally, please correct the classification on line 80: BMI ≥40 kg/m² indicates class III obesity, not class II.
Furthermore, the study sample has not been sufficiently characterized. It remains unclear which types of healthcare workers participated—were they nurses, medical doctors, or other healthcare personnel?
In the Results section, Table 1 would benefit from statistical comparisons between the steatosis and steatosis-free groups (e.g., using a chi-square test). Additionally, numerous biochemical parameters are presented; please ensure these are adequately described in the Methods section.
From line 172 onward, the abbreviation "CAP" appears in both text and tables. Apologies if this was overlooked, but it seems that this abbreviation was not defined previously.
In the Conclusion, an important finding regarding total carbohydrate consumption increasing the odds of hepatic steatosis (Table 4, p = 0.028) has not been mentioned. Please revise.
Finally, Supplementary Table 1 could be moved into the main Results section, as it contains important findings relevant to the study.
Author Response
|
Comments 1: The Introduction section is quite brief. Additional information on the physiological mechanisms underlying the effects of diet on fatty liver should be provided. Furthermore, evidence suggests that high sugar consumption predicts fatty liver, and this should be included. Additionally, it is necessary to clearly outline the novelty and significance of conducting this study specifically among healthcare workers. Why is this particular group important?
|
|
Response 1: Thank you for pointing this out. We expand the introduction by explaining the mechanisms described thus far regarding the pathophysiology of the disease, emphasizing the effect of diet on fatty liver, in addition to including evidence of the effect of excessive sugar consumption on this disease. In short, there are few studies demonstrating that a certain amount and type of sugar directly influences the development of hepatic steatosis (a risk factor); few studies address the maximum amount that should be recommended for treatment, and to date, no clinical practice guidelines address these dietary considerations. Page 2, Paragraph 2, Lines: 47-84. Regarding the population studied, from the protocol proposal, we consider that it is a seldom studied population and although accessible in our environment, it has not been given sufficient importance. In the text that begins on page 2 and continues on page 3, paragraph 3, lines 85-96, we show the studies that have been carried out in similar populations, in which the prevalence of the disease stands out and that despite recognizing what the risk factors are, there is no study that has had as its objective the evaluation of the diet in the disease considering that one or two meals can occur in the hospital where they work.
|
|
Comments 2: Please ensure references are provided to support all study measures, such as bioelectric impedance and dietary assessments. |
|
Response 2: We revised and completed the references about bioelectric impedance on line 155, paragraph 2, page 4; and dietary assessment on line 173, paragraph 1, page 4, thank you.
Comments 3: Additionally, please correct the classification on line 80: BMI ≥40 kg/m² indicates class III obesity, not class II. Response 3: Thank you for your observation, it was a mistake, but now it says class III obesity (corrected on line 134, paragraph 4, page 3).
Comments 4: Furthermore, the study sample has not been sufficiently characterized. It remains unclear which types of healthcare workers participated—were they nurses, medical doctors, or other healthcare personnel? Response 4: We appreciate your comment. We had incorrectly omitted such important data and now describe in the methods section (line 104, paragraph 3, page 3) and results. We have attached more information on the characteristics of the study sample in Table 1. The attached information covers energy expenditure due to physical activity, energy balance, BMI classification categories, and types of healthcare workers. Regarding the types of healthcare workers, the sampled workers were classified into five categories: 1. Medical Staff 2. Nursing Staff 3. Non-Medical Clinical Care Staff 4. Administrative Staff 5. Not specified The fifth category was created as workers were given the option to omit their job for privacy issues due to a slight risk of identification by the research staff, especially for uncommon job tasks.
Comments 5: In the Results section, Table 1 would benefit from statistical comparisons between the steatosis and steatosis-free groups (e.g., using a chi-square test). Additionally, numerous biochemical parameters are presented; please ensure these are adequately described in the Methods section. Response 5: Thank you very much for the suggestion. We felt that comparisons weren't necessary to avoid false findings that could confuse readers, as this is only a descriptive table. On the other hand, we didn't want to make excessive comparisons that would increase the risk of a False Discovery Rate. Despite this, we felt it might be useful for readers to know what possible differences between the groups could suggest possible future hypotheses, so the values were included in Table 1. (between lines 238 y 239, pages 5 to 6). Regarding the biochemical parameters, we attach a few lines in the section “Demographic, Clinical, and Biochemical Data” where it is emphasized that all parameters were obtained through the Beckman Coulter equipment (hematological DxH 1061 and series AU5800 for blood chemistry).
Comments 6: From line 172 onward, the abbreviation "CAP" appears in both text and tables. Apologies if this was overlooked, but it seems that this abbreviation was not defined previously. Response 6: Thank you for pointing out this mistake, we amend because it must be CAP. We rectified the abbreviation and the cut off point according to reference on line 168 paragraph 4, page 4.
Comments 7: In the Conclusion, an important finding regarding total carbohydrate consumption increasing the odds of hepatic steatosis (Table 4, p = 0.028) has not been mentioned. Please revise. Response 7: Thank you for your feedback. We've revised the conclusion text to highlight our results. Table 4 was converted into a graph between lines 303 and 304 on page 12; however, we believe the conclusion should retain its usual structure for reporting main results. |
|
Comments 8: Finally, Supplementary Table 1 could be moved into the main Results section, as it contains important findings relevant to the study. Response 8: Thank you for the suggestion; however, we believe the table is better used as supplementary material, as these results are derived from a collapse of continuous quantitative variables into categories, resulting in the loss of contrast of the continuous variable, which tends to underestimate the effects and possibly induce bias due to inadequate covariate adjustment. The creation of quartiles for dietary intake variables was considered within the study protocol in order to properly explore the various methods suggested by nutritional epidemiology, and for the possible comparison among studies. We would like readers to focus on the results obtained using maximum likelihood methods, in which we use the variables in their original distribution as continuous quantitative variables, while allowing them to consult the table in the supplement for the aforementioned reasons.
|
|
5. Additional clarifications |
|
None |
Reviewer 2 Report
Comments and Suggestions for Authors
Well conducted study and the results are meaningful and the overall manuscript is good.
Author Response
Thank you very much for your revision, we made some changes suggested by other reviewers and we hope you can see them.
Reviewer 3 Report
Comments and Suggestions for Authors
In this article, the authors attempt to investigate the direct relationship between the dietary intake of carbohydrates, fats, and sugars and the composition of the liver and body fat. This study is also relevant and timely, but I would like to suggest to the authors some points that need to be addressed to increase scientific rigour.
In this form, the introduction does not establish sufficient context for the study.
In the Introduction Section, I would suggest that the authors make a broader discussion of the mechanisms by which carbohydrates and fats contribute mechanistically to the development of MASLD.
At the same time, it is important that the authors identify and clearly indicate the existing knowledge gap in this area and how their research aims to address this gap
A reference to the latest dietary guidelines and recommendations would be interesting to better understand the context.
In lines 60-69, I would suggest that the authors also introduce clear explanations for inclusion and exclusion criteria, using proper referencing
Did the researchers also assess physical activity? If so, it should be mentioned; if not, it should be justified why. This is an important factor.
Regarding the BIA measurements, did the participants fast? That is, were the measurements taken at a certain time of the day? These details should be mentioned
I would suggest that the authors introduce some form of visual representation of the results. The way they are presented now, the results are quite difficult to visualise.
Regarding the discussion section I would suggest an extensive reorganisation of it. The discussion section is intended for interpretation of the results. Comparisons should be made with other studies in the literature and possible mechanistic explanations of the results should be offered, based on the literature research. In this form, the discussion section is insufficient.
I suggest that the authors reorganise the discussion so that it follows the logic in which the results were presented. The most important results must draw some conclusions and give possible mechanistic explanations.
Indeed, the authors have correctly pointed out the limitations of the study, but these should be drawn at the end of the section.
For example, I believe that the result on lipid intake and its association with liver fat should be analysed in much more detail, considering that it is a central result of the study.
The average calorie intake is extremely low for a cohort in which a high proportion of participants are obese. This should be explored in depth.
The authors also need to consider metabolic variability.
Comments on the Quality of English Language
The English can be improved.
Author Response
|
Comments 1: In this form, the introduction does not establish sufficient context for the study. In the Introduction Section, I would suggest that the authors make a broader discussion of the mechanisms by which carbohydrates and fats contribute mechanistically to the development of MASLD. At the same time, it is important that the authors identify and clearly indicate the existing knowledge gap in this area and how their research aims to address this gap A reference to the latest dietary guidelines and recommendations would be interesting to better understand the context. Response 1: Thank you for your feedback. We've added some valuable information to the introduction to help you understand the context of our study. From lines 47 to 60 on page 2, we include a concise explanation that explains the mechanism of dietary components in the pathophysiology, including the currently accepted hypothesis for the development of the disease. From this review, we were able to clearly identify the knowledge gap in this area, which ranges from the lack of a determination of the amount of carbohydrates in routine consumption that can be associated with liver damage. Our objective was precisely to add information that allows us to identify it as a risk factor and, in turn, provide data for the treatment of the disease. Lines 66-84, paragraph 1, page 2. We also added the latest recommendations from the main medical associations that address this disease, which are still not referenced in the two senses we discussed. Therefore, we believe that our study can provide valuable information, especially in determining the amount that can increase the likelihood of developing the disease in humans. We were able to find only one study cited in the European guidelines concluding that consuming four servings of sugary drinks (soft drinks) per week increases the likelihood of fatty liver disease. Lines 62 to 67, paragraph 1, page 2.
Comments 2: In lines 60-69, I would suggest that the authors also introduce clear explanations for inclusion and exclusion criteria, using proper referencing Response 2: Thank you, we have further elaborated on the selection criteria, as reflected on line 106 to 109 on paragraph 2, page 3. Specifically, participants were confirmed to be free of specified metabolic disorders, exhibited no evidence of hepatic disease, and provided informed consent for voluntary participation.
Comments 3: Did the researchers also assess physical activity? If so, it should be mentioned; if not, it should be justified why. This is an important factor. Response 3: We agree with this important observation, thank you. On lines 139- 150, pages 3- 4, we describe the questionnaire to collect information about physical activity (between lines 251- 252, page 7. The questionnaire allowed us to obtain the energy expenditure in Kcal for various activities. The results are attached in Table 1, which shows the Kcal values obtained from the questionnaire, as well as their values in transformations into Z scores. Since there is no physical activity classification using the Laval questionnaire to date, the Kcal values were transformed into Z scores and the data for subjects classified as -1 SD, within -1 to 1 SD, and greater than 1 SD are presented. For a better understanding by the reader, the energy balance of the individuals was calculated as the difference between consumption in Kcal and the expenditure in Kcal reported by the Laval questionnaire. The results are presented in the same Table 1, and their transformations to Z scores were also calculated, presenting the same classification of subjects with -1 SD, between -1 and 1 SD, and those greater than 1 SD.
Comments 4: Regarding the BIA measurements, did the participants fast? That is, were the measurements taken at a certain time of the day? These details should be mentioned Response 4: Yes, they fasted 8 hours before the evaluations and blood draw in the morning. We now mention this on line 109- 113, paragraph 2, page 3.
Comments 5: I would suggest that the authors introduce some form of visual representation of the results. The way they are presented now, the results are quite difficult to visualise. Response 5: Thank you very much for this suggestion. We made changes to the manuscript by including the results presented in Table 4 as a combined figure with two Forest plots, which present the results of the multivariate logistic regression model to estimate the probability of fatty liver disease according to the various nutrients studied. We believe this provides a more visual way to understand the results and also maintains important information regarding the ORs, 95% CI, and p values, page 12, between lines 303- 304. Furthermore, we decided to add a new figure presenting the estimates of the probability of having fatty liver disease based on carbohydrate intake per gram and per 15-gram serving. This graph provides readers with more information, showing how the probability increases with increasing intake of the variable that obtained significant results in the multivariate logistic regression model. Page 13, between lines 331- 332.
Comments 6: Regarding the discussion section I would suggest an extensive reorganisation of it. The discussion section is intended for interpretation of the results. Comparisons should be made with other studies in the literature and possible mechanistic explanations of the results should be offered, based on the literature research. In this form, the discussion section is insufficient. I suggest that the authors reorganise the discussion so that it follows the logic in which the results were presented. The most important results must draw some conclusions and give possible mechanistic explanations. Indeed, the authors have correctly pointed out the limitations of the study, but these should be drawn at the end of the section. Response 6: We acknowledge and appreciate this recommendation. The discussion section has been substantially revised in accordance with your suggestions. Indeed, a comprehensive restructuring of this section has been implemented, as evidenced by such modifications. Pages 13 to 16, lines 345- 442.
Comments 7: For example, I believe that the result on lipid intake and its association with liver fat should be analysed in much more detail, considering that it is a central result of the study Response 7: Thank you very much for this suggestion. We made changes to the manuscript to give greater emphasis to our findings. Regarding lipids, a change was made to the conclusion, specifying the result obtained regarding the association with body fat. Furthermore, we added polyunsaturated and monounsaturated lipids to the regression analyses (Table 1, page 10). Although they were not initially part of the study's hypotheses, we believe it could serve to further deepen the information on lipid associations. Primarily because they could have been considered protective factors based on holistic notions of diet. (1).
1.- Alisi A, Agostoni C, Nobili V. Supplementation of monounsaturated and polyunsaturated fatty acids in non-alcoholic fatty liver disease and metabolic syndrome. Lipids. 2011 May;46(5):389-90. doi: 10.1007/s11745-011-3544-2. Epub 2011 Mar 16. PMID: 21409429.
Comments 8: The average calorie intake is extremely low for a cohort in which a high proportion of participants are obese. This should be explored in depth. The authors also need to consider metabolic variability. Response 8: Thank you very much for this observation. We believe one of the reasons for the low caloric intake in our findings is due to the use of a single 24-hour recall. It has been considered standard in nutritional epidemiology to require at least four recalls to obtain a good estimate of dietary intake (1). During the planning of this study, administering more than one 24-hour recall was considered; however, there were insufficient human resources to do so, given the large sample size. There is previous evidence of the use of a single 24-hour recall for dietary estimates in large-sample epidemiological studies. A relationship between BMI and underreporting has been observed, with the finding that as BMI increases, the reporting of fat and energy intake decreases (2). We have added a paragraph to the discussion mentioning this limitation in the study. In the current manuscript, we acknowledge this as a limitation of both the instrument used to collect dietary information and the response bias that participants may have had, and we reflect this in the discussion. Regarding dietary energy, we clarify biases and limitations in lines 415 to 425, paragraph 4, page 15; and we address metabolic variability in lines 408 to 414, paragraph 3, page 15.
1.- Satija A, Yu E, Willett WC, Hu FB. Understanding nutritional epidemiology and its role in policy. Adv Nutr. 2015 Jan 15;6(1):5-18. doi: 10.3945/an.114.007492. PMID: 25593140; PMCID: PMC4288279. 2.- Freedman LS, Commins JM, Moler JE, Willett W, Tinker LF, Subar AF, Spiegelman D, Rhodes D, Potischman N, Neuhouser ML, Moshfegh AJ, Kipnis V, Arab L, Prentice RL. Pooled results from 5 validation studies of dietary self-report instruments using recovery biomarkers for potassium and sodium intake. Am J Epidemiol. 2015 Apr 1;181(7):473-87. doi: 10.1093/aje/kwu325. Epub 2015 Mar 18. PMID: 25787264; PMCID: PMC4371766. |
|
|
|
|
|
4. Comments on the Quality of English Language
|
|
The appraisal of reviewers 1 and 2 is that quality of English is good and does not require improvement, whereas reviewer 3 mentioned that English could be improved to more clearly express the research. Therefore, we conducted extensive review of English grammar and style to guarantee that the manuscript meets the language standards for academic publication. |
|
5. Additional clarifications |
|
None |
Round 2
Reviewer 1 Report
Comments and Suggestions for Authors
Dear Authors,
Thank you for the great job!